# Oxidative Stress-Induced DNA Damage and Apoptosis in Clove Buds-Treated MCF-7 Cells

**DOI:** 10.3390/biom10010139

**Published:** 2020-01-14

**Authors:** Martin Kello, Peter Takac, Peter Kubatka, Tomas Kuruc, Klaudia Petrova, Jan Mojzis

**Affiliations:** 1Department of Pharmacology, Faculty of Medicine, P. J. Safarik University, 040 11 Košice, Slovakia; peter.takac@uvlf.sk (P.T.); tomas.kuruc@student.upjs.sk (T.K.); klaudia.petrova@student.upjs.sk (K.P.); 2Institute of Human and Clinical Pharmacology, University of Veterinary Medicine and Pharmacy, 041 81 Košice, Slovakia; 3Department of Medical Biology, Jessenius Faculty of Medicine, Comenius University in Bratislava, 036 01 Martin, Slovakia; peter.kubatka@uniba.sk; 4Division of Oncology, Biomedical Center Martin, Jessenius Faculty of Medicine, Comenius University in Bratislava, 036 01 Martin, Slovakia

**Keywords:** clove buds extract, apoptosis, breast cancer, oxidative stress, DNA damage, stress/survival pathways

## Abstract

In recent decades, several spices have been studied for their potential in the prevention and treatment of cancer. It is documented that spices have antioxidant, anti-inflammatory, immunomodulatory, and anticancer effects. The main mechanisms of spices action included apoptosis induction, proliferation, migration and invasion of tumour inhibition, and sensitization of tumours to radiotherapy and chemotherapy. In this study, the ability of clove buds extract (CBE) to induce oxidative stress, DNA damage, and stress/survival/apoptotic pathways modulation were analysed in MCF-7 cells. We demonstrated that CBE treatment induced intrinsic caspase-dependent cell death associated with increased oxidative stress mediated by oxygen and nitrogen radicals. We showed also the CBE-mediated release of mitochondrial pro-apoptotic factors, signalling of oxidative stress-mediated DNA damage with modulation of cell antioxidant SOD (superoxide dismutase) system, and modulation activity of the Akt, p38 MAPK, JNK and Erk 1/2 pathways.

## 1. Introduction

Clove, a spice obtained from the dried flower of the tree *Syzygium aromaticum* L., has been used in traditional medicine for centuries as an antiseptic, analgesic, or in dental care [1]. Nowadays, several studies document a broad range of biological effects of clove including antibacterial [2,3], antifungal [4], antimutagenic [5], antihistaminic [6], antiinflammatory [7], or antioxidant [8]. Moreover, the antiproliferative activity of clove essential oil or eugenol, its main component, has also been described [9,10]. The study of Liu et al. [11] showed the antiproliferative effect of clove extract against different types of cancer cells including breast, liver, ovarian or cervical. Recently, the anticancer effect of clove buds extract (CBE) was documented in a model of chemically-induced mammary carcinogenesis [12]. The above mentioned paper described a dose-dependent decrease in tumour frequency by 47.5% and 58.5% when compared to control. Moreover, in particular in vitro experiments, a significant pro-apoptotic effect of CBE has been found associated with cell cycle arrest in S phase, mitochondrial dysfunction, and apoptosis induction.

As mentioned above, cloves possess a strong antioxidant effect. In 2010, Pérez-Jiménez and co-authors [13] published list of the 100 richest dietary sources of phenolic compounds. Among them, clove has been detected as the spice with the highest content of phenolic phytochemicals. In addition, it has been clearly indicated that antioxidant effect of clove is in good correlation with polyphenols content [14]. Furthermore, antioxidant activity of clove extract has been described in several in vitro or in vivo studies [15,16]. On the other hand, although polyphenols are considered antioxidants, some experimental results indicate a possible pro-oxidant effect of these phytochemicals [17,18]. In addition, the results of a recent study also showed a key role of reactive oxygen species in the pro-apoptotic effect of fruit peel polyphenolic extract [19]. Furthermore, the involvement of oxidative stress in the antibacterial activity of clove seeds extract has also been described [20].

These results prompted us to investigate whether the pro-apoptotic effect of CBE can be associated with the induction of oxidative stress in cancer cells. Moreover, our attention was also focused on different signalling pathways (JNK, MAPK, Erk1/2, Akt) and DNA damage signalling in CBE-treated cells.

## 2. Materials and Methods

### 2.1. Cell Cultures and CBE Treatment

The human cancer cell line MCF-7 (human breast adenocarcinoma) and MCF-10A (human mammary gland epithelial cells) were obtained from ATCC- American Type Culture Collection (Manassas, VA, USA). MCF-7 cells were cultured in a DMEM medium with sodium pyruvate (GE Healthcare, Piscataway, NJ, USA) and MCF-10A in growth medium consisting of high glucose DMEM F12 Medium (Biosera, Kansas City, MO, USA) + Supplement additives (insulin, EGF- epitelial growth factor, HC-hydrocortisone, choleratoxin- all Sigma, Steinheim, Germany). The growth medium was supplemented with a 10% fetal bovine serum (FBS) (Invitrogen, Carlsbad, CA, USA) and 1X HyClone™ Antibiotic/Antimycotic Solution (GE Healthcare, Piscataway, NJ, USA). Cells were maintained in standard cancer cell culture conditions (5% CO_2_ in humidified air at 37 °C). Cell viability before all experiments was greater than 95%.

CBE ethanol extract was purchased commercially from company Calendula (Nová Ľubovňa, Slovak Republic) as a 40% ethanol solution. The final CBE content was 971 mg/mL. The extract was registered under batch number S-01-01-12-02-15.

Cells were treated with CBE ethanol extract for 1–72 h prior to analysis. Ethanol final concentration in experimental groups containing CBE extract (c = 350 resp. 450 μg/mL) experiments was max. 0.4% with no toxicity. Secondary metabolites identification in CBE extract was performed and characterised as was described in previous paper [12].

### 2.2. Reagents

Reagents used in experiments: MTS (Sigma-Aldrich Chemie, Steinheim, Germany); Cytochrome c Antibody (6H2) FITC Conjugate (Invitrogen, Carlsbad, CA, USA); Smac/Diablo Rabbit mAb (Cell Signalling Technology^®^, Danvers, MA, USA); Cleaved Caspase-9 (Asp315) Rabbit mAb PE conjugate (Cell Signalling Technology^®^, Danvers, MA, USA); Anti-Bad antibody DyLight^®^ 488 (Abcam, Cambridge, UK); Phospho-Bad (Ser112) Rabbit mAb PE Conjugate (Cell Signalling Technology^®^, Danvers, MA, USA), Goat anti-rabbit IgG (H + L) Secondary Antibody Alexa Fluor 488 (Thermo Fisher Scientific, Rockford, IL, USA); MitoSOXTMRed mitochondrial superoxide indicator (Thermo Fisher Scientific, Rockford, IL, USA), dihydrorhodamine-123 (DHR-123, Sigma-Aldrich, Steinheim, Germany); DAF-FM diacetate (Diaminofluorescein-FM, Sigma-Aldrich, Steinheim, Germany); BODIPY 581/591 C11(Thermo Fisher Scientific, Rockford, IL, USA); Anti-pATM, PE Conjugated Antibody; Anti-pHistone H2A.X, PerCP Conjugated Antibody; Anti-pSMC1, Alexa Fluor^®^ 488 Antibody (all Millipore Corporation, Temecula, CA, USA); anti-p53 (1C12) Mouse mAb Alexa Fluor^®^ 488 Conjugate and Phospho-p53 (Ser15) Mouse mAb PE Conjugate; Phospho-SAPK/JNK (Thr183/Tyr185) (G9) Mouse mAb PE Conjugate (all Cell Signalling Technology^®^, Danvers, MA, USA); Recombinant Anti-SOD2/MnSOD antibody (Abcam, Cambridge, UK); Western blot primary and secondary antibody from Table 1 (Cell Signalling Technology^®^, Danvers, MA, USA); Pierce^®^ BCA Protein Assay Kit (Thermo Fisher Scientific, Rockford, IL, USA); chemiluminescent ECL substrate (Thermo Fisher Scientific, Rockford, IL, USA).

### 2.3. Viability Test

The MTS colorimetric test [21] was used to determine the antiproliferative effect of CBE and N-acetylcysteine. MCF-7 cells (1 × 10^4^/well) were seeded in 96-well polystyrene microplates (SARSTEDT, Nümbrecht, Germany). Twenty-four hours after seeding CBE final concentrations 350 and 450 μg/mL and NAC (final concentration 1.5 mM) or Z-VAD-FM (50 µM as pre-treatment; Enzo Life Sciences, Inc., Farmingdale, NY, USA) or their combinations were added. After 72 h cells were incubated with 10 μL of MTS (5 mg/mL) at 37 °C. After an additional 2 h, cell proliferation was evaluated by measuring the absorbance at wavelength 490 nm using the automated Cytation™ 3 Cell Imaging Multi-Mode Reader (Biotek, Winooski, VT, USA). Absorbance of control wells was taken as 1.0 = 100%, and the results were expressed as a fold/percentage of untreated control. All experiments were performed in triplicate.

### 2.4. Flow Cytometric Analysis (FCM)

The MCF-7 cells (1 × 10^6^) were seeded for FCM analyses [22] in Petri dishes and treated with CBE at 350 and 450 μg/mL concentrations for 1, 3, 6, 12, 24, 48 or 72 h depending on experimental scheme. Floating and adherent cells were harvested, washed in PBS, divided for particular analysis and stained prior to analysis (as described in Section 2.4.1, Section 2.4.2, Section 2.4.3). Fluorescence was detected after 15–30 min incubation at room temperature in the dark using a BD FACSCalibur flow cytometer (Becton Dickinson, San Jose, CA, USA). A minimum of 1 × 10^4^ events were analysed per analysis. All experiments were performed in triplicate.

#### 2.4.1. Detection of Mitochondrial Apoptotic Pathway Associated Proteins

Cytochrome *c* release, Smac/DIABLO accumulation, caspase-9 activity and Bad release/phosphorylation status were analysed with FCM using Cytochrome *c* Antibody (6H2) FITC Conjugate; Smac/Diablo Rabbit mAb + Goat anti-rabbit IgG (H + L) Secondary Antibody Alexa Fluor 488; Cleaved Caspase-9 (Asp315) Rabbit mAb PE conjugate; Anti-Bad antibody DyLight^®^ 488 and Phospho-Bad (Ser112) Rabbit mAb PE Conjugate. The MCF-7 cells were harvested 24, 48 and 72 h after CBE treatment (350 and 450 μg/mL). Cell population was stained with conjugated antibody and incubated for 30 min at room temperature in the dark or stained with primary antibody (30 min), followed by secondary conjugated antibody staining (15 min in dark). The cells were then washed with PBS, resuspended in 500 μL of the total volume in PBS, and analysed (1 × 10^4^ cell per sample) by a BD FACSCalibur flow cytometer. Detection was performed in triplicate.

#### 2.4.2. Measurement of Reactive Oxygen/Nitrogen (ROS, RNS) Species Accumulation and Lipid Peroxidation

Oxygen/nitrogen radicals are produced intracellularly and detected with FCM analysis using MitoSOXTMRed mitochondrial superoxide indicator, dihydrorhodamine-123 (DHR-123), which reacts with intracellular hydrogen peroxide (ROS), DAF-FM (Diaminofluorescein-FM) diacetate, reacting with nitrogen radicals and BODIPY 581/591 C11, detecting lipid peroxides. The cells treated with CBE (350 and 450 μg/mL) were harvested 1–72 h after treatment, washed two times in PBS and resuspended in PBS. DHR-123 was added at a final concentration 0.2 µM, MitoSOX red at 5 µM, DAF/FM DA at 2 mM and BODIPY at final concentration 1 µM. The samples were then incubated for 15–30 min in the dark at room temperature and after incubation were placed on ice. Fluorescence was detected by flow cytometer (BD FACSCalibur) with optical filters: 530/30 BP (FL-1) for DHR-123, 585/42 BP (FL-2) for DAF-FM or 670 LP (FL-3) for BODIPY. Forward and side scatters were used to gate the viable populations of cells. Detection was performed in triplicate.

#### 2.4.3. Measurement of DNA Damage-Associated Proteins

Damage to DNA strands was detected by phosphorylation of histone H2A.X and kinases ATM, SMC1 and by activation of protein p53 as major regulator of DNA repair mechanisms. The MCF-7 cells treated with CBE (350 and 450 μg/mL) were harvested 24, 48, and 72 h after treatment, washed two times in PBS and resuspended in PBS. Cell suspensions were stained with Anti-pATM, PE Conjugated Antibody; Anti-pHistone H2A.X, PerCP Conjugated Antibody; Anti-pSMC1, Alexa Fluor^®^ 488 Antibody; anti-p53 (1C12) Mouse mAb Alexa Fluor^®^ 488 Conjugate and Phospho-p53 (Ser15) Mouse mAb PE Conjugate 15–30 min in the dark at room temperature and analysed by BD FACSCalibur flow cytometer. Detection was performed in triplicate.

### 2.5. Western Blot Analysis

The Laemli lysis buffer containing 1 mol/L Tris/HCl (pH 6.8), glycerol, 20% SDS (sodium dodecyl sulphate) and deionized H_2_O in the presence of PIC (protease inhibitor cocktail, Sigma-Aldrich Chemie, Steinheim, Germany) and a sonication process was used to prepare protein lysates from the MCF-7 cells after treatment. The concentration of proteins was quantified using the Pierce^®^ BCA Protein Assay Kit and measured using an automated Cytation™ 3 Cell Imaging Multi-Mode Reader (Biotek, Winooski, VT, USA) at wavelength 570 nm. Proteins were separated on SDS-PAA gel (12%) at 100 V for 2 h and then transferred to a PVDF Blotting Membrane (GE Healthcare, Piscataway, NJ, USA) at 200 mA for 2 h using a BioRad Mini Trans-Blot cell (BioRad, Hercules, CA, USA). To minimize nonspecific binding, the membrane with the transferred proteins was blocked in 4% milk with TBS-Tween (pH 7.4) for 1 h at room temperature. The incubation of membrane with primary antibodies was subsequently set overnight at 4 °C. Immunoblotting was carried out with the antibodies stated below (Table 1). After incubation process, the membranes were washed in TBS-Tween (3 × 5 min) and incubated with a corresponding secondary antibody associated with horseradish peroxidase for 1 h at room temperature. Afterward, the membranes were again washed in TBS-Tween (3 × 5 min), and the expression of proteins was detected using a chemiluminescent ECL substrate and MF-ChemiBIS 2.0 Imaging System (DNR Bio-Imaging Systems, Jerusalem, Israel) [23]. Densitometry analysis of proteins bands intensity was performed in Image Studio Lite software (LI-COR Biosciences, Lincoln, NE, USA) using β-actin as loading protein control. Untreated cell lysates were afterwards used to normalise all experimental groups. Detection was performed in triplicate.

#### Detection of Stress/Survival Proteins

Flow cytometry and Western blot analyses of total and phosphorylated proteins involved in stress/survival pathways were performed. The MCF-7 cells treated with CBE (350 and 450 μg/mL) were harvested 24, 48 and 72 h after treatment, washed two times in PBS and stained 15–30 min with Phospho-SAPK/JNK (Thr183/Tyr185) (G9) Mouse mAb (PE Conjugate) at room temperature or lysed for western blot analyses. In western blot analyses, proteins from MCF-7 cells were stained as described in Table 1. Detection was performed in triplicate.

### 2.6. Statistical Analysis

Results are expressed as mean ± SD. Statistical analysis of the results were performed using standard procedures, with one-way ANOVA followed by the Bonferroni multiple comparisons test. Data analyses were conducted using GRAPHPAD PRISM, version 5.01(GraphPad Software, La Jolla, CA, USA). Values of *p* < 0.05 were considered to be statistically significant. Throughout this paper, * indicates *p* < 0.05, ** *p* < 0.01, *** *p* < 0.001.

## 3. Results

In our previous paper [12] we described anti-cancer activity of CBE extract in breast cancer model in vitro and in vivo. The results obtained pointed out to significant pro-apoptotic, antiproliferative and anti-angiogenic effects of CBE in mammary carcinoma model. However, the mechanisms of pro-apoptotic activity of CBE have not been sufficiently described.

### 3.1. Effect of NAC and Z-VAD-FM on CBE-Induced Cytotoxicity

Clove buds extract significantly decreased cell survival in both concentrations after 72 h of incubation. Co-incubation of CBE with NAC significantly prevented decrease in CBE-treated cell survival of CBE indicating involvement of ROS in cytotoxicity of CBE. Furthermore, co-treatment of CBE with pancaspase inhibitor Z-VAD-FM also prevented CBE-induced cytotoxicity suggesting caspase-dependent cell death (Figure 1). In addition, the cytotoxicity of CBE in non-cancer MCF-10A cells was negligible (Appendix A).

### 3.2. Effect of CBE Treatment on Intrinsic Apoptotic Pathway Alterations

To support findings that CBE treatment induced apoptosis in MCF-7 cells, herein, testing involved the mitochondrial intrinsic apoptotic pathway. As the results showed, CBE treatment with both concentrations induced significant release of cytochrome *c* soon after 24 h of incubation (Figure 2A). Moreover, cytochrome *c* release and increased expression of Smac/DIABLO (Figure 2B) after 48 and 72 h together led to significant caspase-9 activation (Figure 2C). Besides that, increased levels of released pro-apoptotic Bad proteins from mitochondria (Figure 2D) after CBE treatment and decreased phosphorylation status of Bad (Figure 2E) after 48 and 72 h, supported intrinsic apoptotic pathway alterations by CBE extract.

### 3.3. Oxidative Stress Induction Mediated by CBE Treatment

Several mechanisms affected programmed cell death, include oxidative stress, can be mediated and triggered by various treatments. In general, oxygen and nitrogen radicals are involved in DNA and organelle damage and act as inductors of apoptotic process. The three different species of oxygen radicals were analysed: superoxide anions (Figure 3A), ROS (peroxides intermediates) (Figure 3B), lipids peroxides (Figure 3C) and moreover, also relative levels of intracellular nitrogen radicals (RNS) (Figure 3D) were determined. Despite the scavenging activity of CBE treatment after 1–6 h, we noticed two major oxygen species formation after CBE administration from 12 resp. 24 h, with culmination at 48 h (O^2−^) resp. 72 h (peroxides), which contributed probably to several pro-apoptotic processes. Moreover, increased lipid peroxide formation after 12 h and a significant time-dependent increase of RNS soon after 1 h suggested complex oxidative stress induction after CBE treatment.

### 3.4. CBE-Mediated DNA Damage and Antioxidant System Alterations

Oxidative stress mediated by all kind of oxygen and nitrogen radicals usually leads to activation of intracellular antioxidant self-defence (SOD, catalase, etc.). Balance between radical’s generation and scavenging activity of cellular antioxidant enzymes decide about cells destiny favouring survival or programmed cell death after DNA damage. As the results showed, DNA damage induced activation of ATM (Ataxia-telangiectasia mutated) kinase as response to double-strand breaks occurred at 24–72 h after CBE treatment (Figure 4B). Moreover, DNA damage also activated cell cycle checkpoint protein p53 expression (Figure 4D) at 24 h. Activation of ATM kinase subsequently phosphorylated downstream factors, included p53 (Figure 4E), SMC1 (Figure 4C), and histone variant H2A.X (Figure 4A) mostly at 48–72 h after treatment. Moreover, we noticed, that long-term (48–72 h) and higher concentration (450 μg/mL) CBE treatment led to increased expression of both SOD enzymes (Figure 5A,C): SOD 1 (48–72 h) and SOD 2 (72 h) due to higher radicals burst in MCF-7 cells. A lower concentration (350 μg/mL) of CBE significantly inhibited the expression of SOD 1 at 24–48 h and SOD 2 completely at 24 to 72 h. However, despite the effort of the SOD system to protect MCF-7 cells from oxidative/nitrogen stress, we still noticed DNA damage signalling.

### 3.5. Effect of CBE Treatment on MAPK, Erk, Akt and JNK Activation

Protein members of cell stress/survival pathways play a crucial role in the response of cells to external and internal stimuli. Therefore, we studied the effect of CBE on MAPK protein family members. We detected phosphorylation changes and the activity modulation of Erk 1/2 (extracellular signal-regulated kinases), JNKs (c-Jun *N*-terminal kinases), Akt, and p38-MAPKs (Figure 5A–C). De-/activation of these proteins affect several processes, including cell growth, differentiation, development, apoptosis, and cell survival. We noticed time-dependent and concentration-dependent regulation (degradation) of total Erk 1/2 and Akt, but not total p38 MAPK kinases after CBE treatment. Activation analyses showed that phosphorylation of p38 MAPK and JNK significantly increased after 48 h after treatment, similar as Erk 1/2 but only in higher concentration. Moreover, phosphorylation of Akt increased after 24 h. The amount of phosphorylated p38 MAPK, Erk 1/2 and Akt significantly decreased time- and concentration dependent at 48 h with minimum at 72 h after CBE treatment. Only JNK phosphorylation culminated at 72 h. It is obvious that CBE treatment is able to modulate stress/survival proteins activity supporting pro-apoptotic potential of CBE extract.

## 4. Discussion

Recently, the anticancer effect of CBE was demonstrated in an in vivo mammary cancer model. Moreover, in vitro experiments showed pro-apoptotic effect of CBE in MCF-7 cells associated with decrease in mitochondrial membrane potential, deactivation of Bcl-2 anti-apoptotic activity as well as activation of caspase-7 [12]. However, the precise mechanisms of antiproliferative activity of CBE remain unknown. In order to better understand how CBE modulates the death pathway in cancer cells, a set of additional experiments focused on (i) the identification of molecular mechanisms of CBE-induced apoptosis and (ii) the involvement of free radicals in CBE-induced cell death were performed.

Bcl-2 family proteins play an important role in mitochondria-mediated apoptosis via the regulation of mitochondrial outer membrane (MOM) permeabilization [24]. To preserve MOM integrity, it is important to restrain pro-apoptotic Bax/Bak by anti-apoptotic proteins of Bcl-2 family. On the other hand, activation of Bax/Bak by pro-apoptotic proteins such as Bim, Bid or Bad resulted in MOM permeabilization [25]. Once MOM is permeabilized, pro-apoptotic proteins such as cytochrome *c* and Smac/DIABLO are released from the intermembrane space into the cytosol, resulted in caspase activation and apoptosis induction [26]. In the present study, a significant increase of total Bad in CBE-treated cells was found. It is broadly accepted that only non-phosphorylated Bad may block anti-apoptotic Bcl-xL and Bcl-2 proteins, release pro-apoptotic Bax and initiate MOM permeability [27]. However, the experiments presented here showed increased levels of phosphorylated Bad after 24 h of treatment with a subsequent decrease of phosphorylation to control levels after 48 and 72 h of incubation (Figure 1 D,E). Despite the absence of direct evidence, we suggest that Bad phosphorylation may be mechanism of cancer cells self-protection. However, because it has been found increased levels of released cytochrome *c* and Smac/DIABLO as well as activation of caspase-9 after 48 and 72 h of treatment, Bad phosphorylation was not able to override apoptotic machinery and cell death. The presented results are in agreement with those described pro-apoptotic effect of CBE or its active constituents via the modulation of activity of Bcl-2 proteins, cytochrome *c* release, or caspase activation [28,29].

Another characteristic aspect of apoptosis is poly (ADP-ribose) polymerase (PARP) cleavage. This nuclear enzyme is involved in several of cellular processes such as regulation of genomic stability, DNA repair, transcription and apoptosis. PARP is the first responder that detects single-strand DNA breaks and start a cellular machinery leading to DNA repair. On the other hand, PARP is also substrate for caspases (e.g., caspase-3 and -7) and its cleavage is an indicator of apoptosis [30]. In the present study, time- and dose-dependent PARP cleavage was found. Because MCF-7 cells do not express caspase-3 [31], we suggest that PARP cleavage can be the result of CBE-induced caspase-7 activation. The data presented here are in accordance with the findings of Liu and co-workers [29] who documented the pro-apoptotic effect of active fraction of clove associated with caspase activation and PARP cleavage.

Furthermore, mitogen-activated protein kinases (MAPK) is another important mechanism in apoptosis control. There are three classical MAPK including extracellular signal-regulated kinases (Erk1/2), c-Jun N-terminal kinase (JNK) and p38-MAPK. Activated MAPK are involved in several cellular processes including cell survival and apoptosis. Originally, it has been shown that Erk1/2 after activation is involved in cell survival [32], while activation of JNK and p38-MAPK have been involved in apoptosis induced by several anticancer agents such as polyphenols [19], thymoquinone [33], cyclophosphamide [34], and indole phytoalexins [35]. In accordance with these studies, a significant increase in phosphorylated JNK and p38-MAPK in CBE-treated cancer cells was observed. However, the result also showed the increased phosphorylation of Erk1/2. Although activation of ERK signalling is considered to be pro-survival, several lines of evidence documented activation of the ERK pathway in DNA-damage induced apoptosis after anticancer drug [36,37] or natural compounds [38,39,40] treatment. In addition, many studies related to apoptosis induced by the ERK pathway documented unusually prolonged ERK activation [41,42]. Furthermore, considerable evidence from experimental studies documents an association among ERK activation, ROS production, and cell death [43,44]. These results are in accordance with findings reported here. The analysed results show a significant increase in phosphorylated Erk1/2 after 48 and 72 h of treatment with CBE (Figure 5A,C), as well as a time and dose-dependent increased production of ROS, lipid peroxides, and reactive nitrogen species (Figure 2). It is clear that eugenol, as part of the clove buds extract, represent a major substance associated with ROS production and lipid peroxidation [45,46]. It is also suggested that ROS can be involved also in activation of the other members of MAPK signalling including JNK and p38-MAPK [47,48,49]. These results are in accordance with obtained data documented increased production of ROS associated with phosphorylation of both JNK and p38 MAPK. To the best of our knowledge, this is the first study to report an association of Erk, JNK, and p38 MAPK in ROS-mediated apoptosis after clove bud extract treatment in a carcinoma model.

In addition to MAPK signalling, the role of ROS in DNA damage has been well documented [50,51]. Furthermore, several articles described a relation between DNA damage (e.g., DNA double-strand breaks (DSB), O^6^-methylguanine, base N-alkylations or DNA cross-links) and apoptosis induction [52,53]. In light of this evidence, the DNA damage in CBE-treated MCF-7 cells has been studied. The consequence of DNA damage is activation of the DDR (DNA damage response) mechanism responsible for DNA damage detection, block of cell cycle, as well as activation of repair machinery resulting in cell survival or possibly apoptosis [53]. One of the first steps in the DDR is the phosphorylation of ataxia telangiectasia-mutated (ATM) and/or ATM and Rad3-related (ATR) kinases with subsequent regulation of downstream targets such as p53, histone H2A.X or structural maintenance of chromosomes (SMC1) protein [54,55]. Phosphorylation (i.e., activation) of these proteins is commonly known as a marker for DSB. The presented results showed a significant increase in ATM, p53 protein, SMC1 protein as well as histone H2A.X (γ-H2A.X) phosphorylation indicating severe CBE-induced DNA damage. Extract from clove as well as its main component, eugenol, have been shown to phosphorylate histone H2A.X to a similar extent either in human colon cancer cell or in Chinese hamster ovary cells [11,56]. On the other hand, the phosphorylation of ATM and SMC1 protein in CBE or eugenol-treated cells, to the best of our knowledge, has not been published yet.

Besides DNA damage, oxidative stress mediated by CBE also induces in MCF-7 cells increasing dose- and time-dependent activation of SOD defence. However, the results showed that SOD system reactivity is attenuated and is not able to prevent increasing oxidative stress leading to apoptosis. Moreover, this fact is also in accordance with study of Al Wafai and co-workers [57], where SOD pre-treatment was not able prevent eugenol-mediated increase of ROS levels in MCF-7 cells.

Through the phosphorylation of various targets, including apoptosis or cell cycle regulators, activated Akt regulates cell-cycle progression, cell apoptosis and survival [58]. De-phosphorylation of Akt, as a result of inhibition in the conversion of PIP2 to PIP3 by PTEN, lead to apoptosis-promoting signalling. As the study showed, CBE treatment, led to the time- and dose-dependent deregulation of Akt in MCF-7 cells. Similarly, eugenol inhibited the progress of breast precancerous lesion by inhibiting phosphorylation of Akt and blocking HER2/PI3K-AKT signalling pathway [59]. Furthermore, the presented data also showed link between Akt and Bad, where decreased phosphorylation of Bad is related to de-phosphorylating Akt mediated by CBE treatment. It is known that unphosphorylated Akt activates the pro-apoptotic actions of BAD [60].

## 5. Conclusions

In conclusion, the presented data show that clove bud extract induced apoptosis in human breast cancer MCF-7 cells. Apoptosis was accompanied with ROS production and DNA damage with the subsequent activation of DNA repair mechanism including phosphorylation of ATM, HA2.X, and SMC1 protein as well as p53 activation. Moreover, CBE treatment led to phosphorylation of MAPK signalling (Erk, JNK, p38 MAPK) and Akt. Furthermore, the modulation of Bcl-2 family proteins and leakage of pro-apoptotic proteins from mitochondria indicated the mitochondrial pathway of apoptosis. A proposed model depicting the molecular mechanism underlying CBE involvement in MCF-7 cell death is presented in Figure 6.

Together with the previously published paper [12], presented results suggest that clove buds could be further developed as an anticancer agent.

## Figures and Tables

**Figure 1 biomolecules-10-00139-f001:**
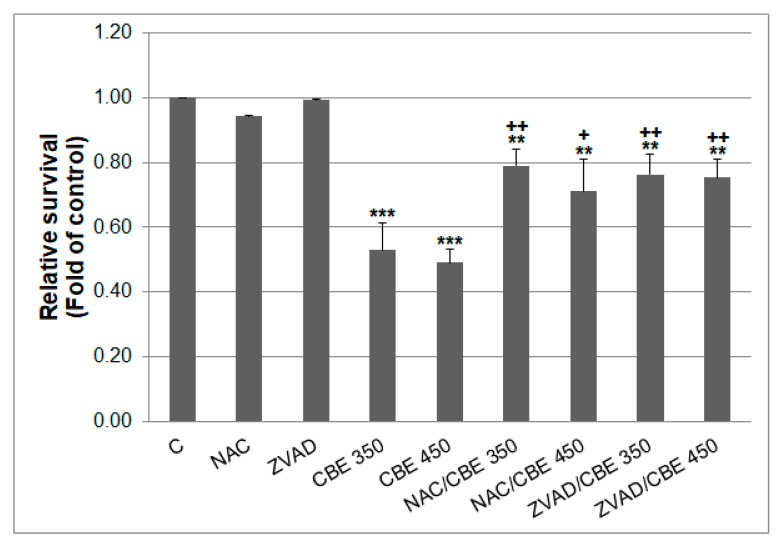
MCF-7 cells proliferation after CBE or NAC/CBE or ZVAD/CBE combination treatment. Data were obtained from three independent measurements. Significantly different ** *p* < 0.01, *** *p* < 0.001 vs. untreated cells (control) and ^+^
*p* < 0.05, ^++^
*p* < 0.01 vs. CBE treatment.

**Figure 2 biomolecules-10-00139-f002:**
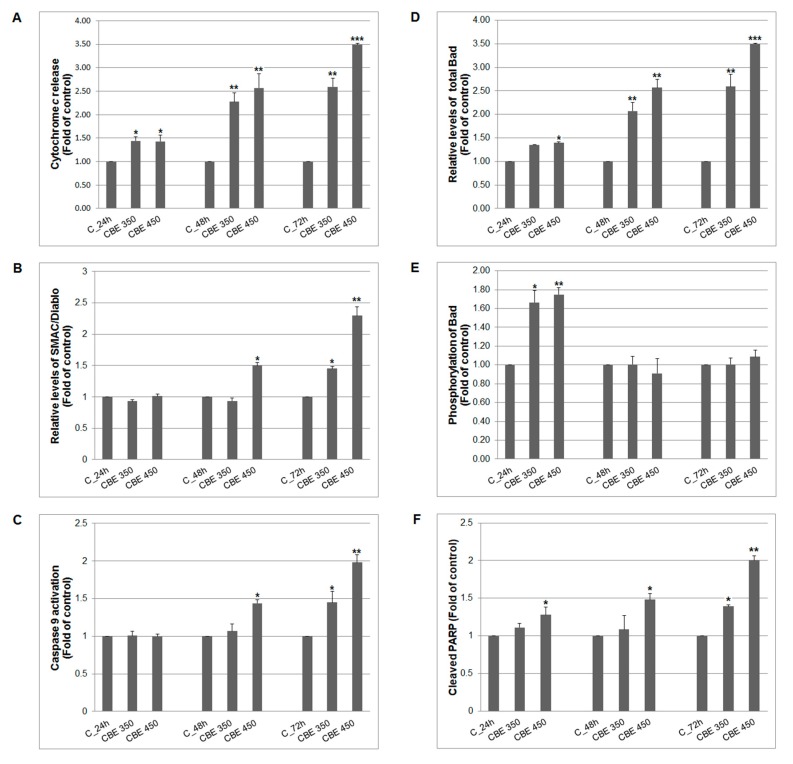
Effect of CBE on (**A**) cytochrome *c* release, (**B**) SMAC/Diablo expression, (**C**) caspase-9 activation, (**D**,**E**) Bad activity and cleaved PARP (**F**) in MCF-7 cells. Data were obtained from three independent flow cytometry experiments after 24, 48 and 72 h of CBE treatment. Significance: * *p* < 0.05, ** *p* < 0.01, *** *p* < 0.001 vs. untreated cells (control).

**Figure 3 biomolecules-10-00139-f003:**
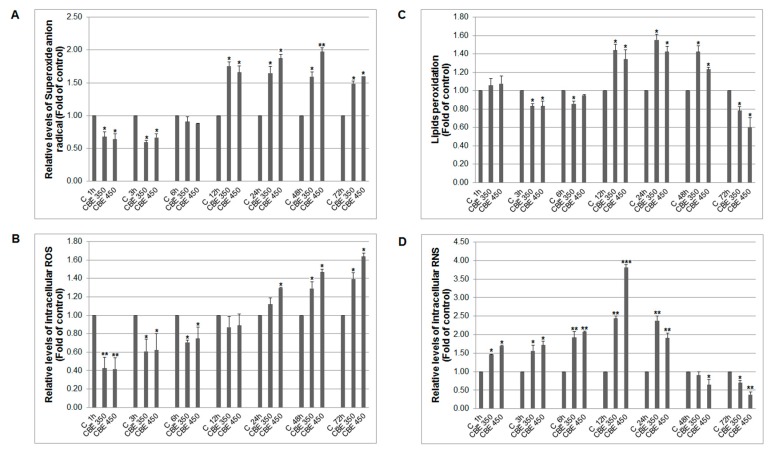
Effect of CBE on relative levels of (**A**) superoxide anions, (**B**) intracellular ROS, (**C**) lipids peroxides and (**D**) intracellular RNS in MCF-7 cells. Data were obtained from three independent flow cytometry experiments after 1–72 h CBE treatment. Significance: * *p* < 0.05, ** *p* < 0.01, *** *p* < 0.001 vs. untreated cells (control).

**Figure 4 biomolecules-10-00139-f004:**
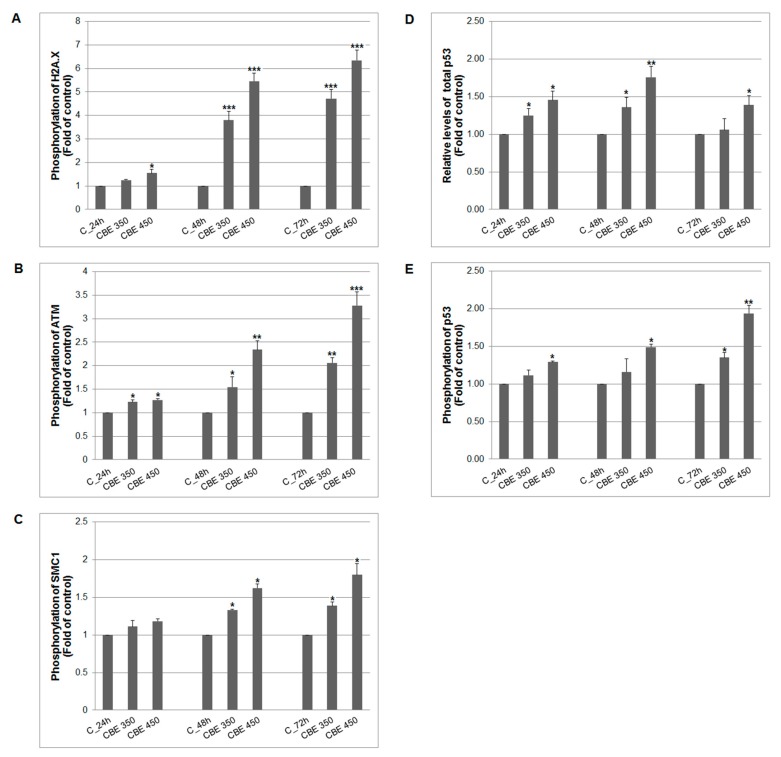
Effect of CBE on phosphorylation status of (**A**) histone H2A.X, (**B**) ATM kinase, (**C**) SMC1 kinase and (**D**,**E**) p53 activity in MCF-7 cells. Data were obtained from three independent flow cytometry experiments after 24, 48 and 72 h CBE treatment. Significance: * *p* < 0.05, ** *p* < 0.01, *** *p* < 0.001 vs. untreated cells (control).

**Figure 5 biomolecules-10-00139-f005:**
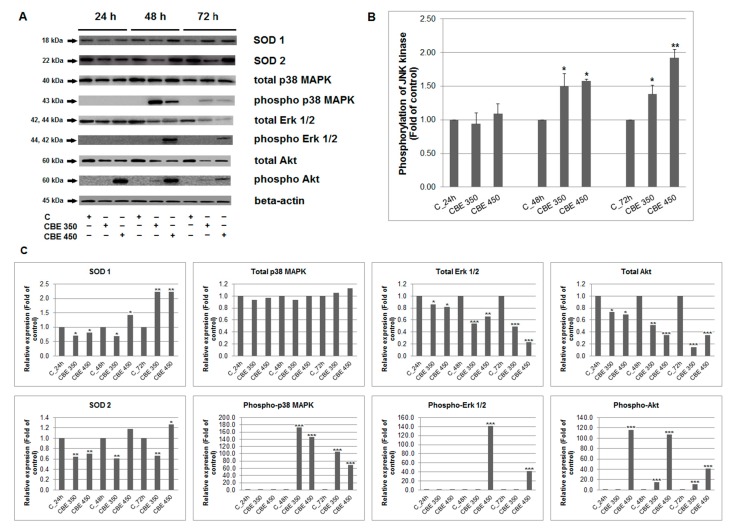
Effect of CBE on (**A**) protein expression, (**B**) JNK kinase activity in MCF-7 cells. (**C**) Densitometry analyses of western blot results. Representative data of three independent experiments after 24, 48 and 72 h CBE treatment are presented. Significance: * *p* < 0.05, ** *p* < 0.01, *** *p* < 0.001 vs. untreated cells (control).

**Figure 6 biomolecules-10-00139-f006:**
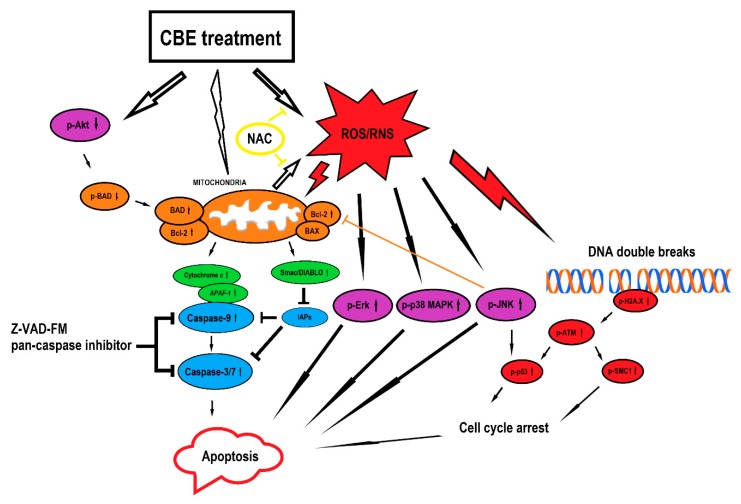
Proposed model of the signaling pathways underlying CBE-induced apoptosis in MCF-7 cells.

**Table 1 biomolecules-10-00139-t001:** List of western blot antibodies.

**Primary Antibodies**	**Mr (kDa)**	**Origin**	**Company**
β-actin	45	Mouse	Cell Signalling Technology^®^
SOD 1	18	Rabbit	
SOD 2	22	Rabbit	Abcam
p38 MAPK	40	Rabbit	Cell Signalling Technology^®^
phospho-p38 MAPK	43	Rabbit	
Akt	60	Rabbit	
phospho-Akt	60	Rabbit	
Erk 1/2	42, 44	Rabbit	
phospho-Erk 1/2	44, 42	Rabbit	
**Secondary Antibodies**		**Origin**	**Company**
Anti-rabbit IgG HRP	=	Goat	Cell Signalling Technology^®^
Anti-mouse IgG HRP	=	Goat	

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
