# Peer review of "Oxidative Stress-Induced DNA Damage and Apoptosis in Clove Buds-Treated MCF-7 Cells"

_biomolecules, 2020, doi:10.3390/biom10010139_

Round 1

Reviewer 1 Report

The manuscript by Kello et al. investigates some aspects of the molecular mechanism of how clove buds extracts induce DNA damage and apoptosis in MCF7.

Although the work is potentially interesting to readers of the journal Biomolecules, nonetheless, the data presented here are preliminary, some results are unclear or not well explained.

In my opinion the authors:

-should evaluate which component of the extracts they use can induce the effects they highlight (they could thus understand if there are pro-oxidant compounds and other antioxidants);

-should evaluate the effects of zVAD and N-acetyl cystein on cell viability. A non-tumor line should also be used as a control (such as HMEC; MCF10);

-should evaluate the effects of N-acetyl cystein on some key factors analyzed by them (erk, JNK, Bad and so on);

-should review the following paragraphs of materials and methods (2.3 and 2.5). Lack of information (it is not clear how the various data were developed, also in 2.4 the optical filters for which analyzes were used respectively should be specified) and the initial description should not be included in the materials and methods. In 2.2 were used 1x106 cells for analisys? In 2.3 line 91....500 uM?

- should specify the meaning of the abbreviation CLO;

-should specify how normalization was performed for western blot analysis;

-should insert in the legend of figure 2 ***;

- should change the title of the results section 2.7 because they do not evaluate the activity

Superoxide anion is a ROS

Author Response

Reviewer 1

In my opinion the authors:

 -should evaluate which component of the extracts they use can induce the effects they highlight (they could thus understand if there are pro-oxidant compounds and other antioxidants);

Composition of extract were detailed analysed (GC-MS and LC-MS-DAD analysis) in our previous paper:

Kubatka, P., Uramova, S., Kello, M., Kajo, K., Kruzliak, P., Mojzis, J., ... & Zubor, P. (2017). Antineoplastic effects of clove buds (Syzygium aromaticum L.) in the model of breast carcinoma. Journal of cellular and molecular medicine, 21(11), 2837-2851.

These analyses confirmed (as other literature data) that major component of clove buds extract were eugenol (235 mg/ml) and the derivatives of caffeoylquinic acid: chlorogenic acid (61 mg/ml) and monocaffeoylquinic acid (64 mg/ml). It was also identified that eugenol have relative percentage content of 80.9% and eugenyl acetate of 5.4%. The rest of volatile compounds were in relative contents less than 1% in whole extract.

 As is known from literature, eugenol can act as dual antioxidant/prooxidant mediator in cancer development and treatment, depended of concentration, treatment duration, cell line tested etc.. Therefore we also discussed our collected data mostly with eugenol or flavonoids/polyphenols compounds.

-should evaluate the effects of zVAD and N-acetyl cystein on cell viability. A non-tumor line should also be used as a control (such as HMEC; MCF10);

We agree and we used non-tumor cell line MCF-10A in MTS preliminary/screening analyses, where only higher concentration showed some toxicity/inhibition (Suppl.1). In our above mention paper we published most related data about cytotoxicity, antiproliferative activity, apoptosis induction of CBE treatment in in vitro model. Furthermore, we published several original data from in vivo experiment. In addition, we added results of experiments with  Z-VAD-FM and NAC (Fig. 1)

-should evaluate the effects of N-acetyl cystein on some key factors analyzed by them (erk, JNK, Bad and so on);

We showed that NAC significantly prevented CBE-induced cytotoxicity. We agree with reviewer, that for identification target molecules, some another factors should be analysed. However, there are time consuming experiments and it is not possible to do it in 10-day period we have for manuscript revision.

-should review the following paragraphs of materials and methods (2.3 and 2.5). Lack of information (it is not clear how the various data were developed, also in 2.4 the optical filters for which analyzes were used respectively should be specified) and the initial description should not be included in the materials and methods.

Corrected, Data about flow cytometer, numbers of repeats and filter specification (2.4) were added. Initial descriptions were removed as you pointed.

In 2.2 were used 1x106 cells for analisys?

Accepted, Methods description was little confused, therefore we corrected it. MCF-7 cells were seeded for flow cytometry analyses in density 1x106, treated with CBE and analysed in specific time point as descripted. Because cells were divided to particular analyses with different antibody used, the minimal cell count was performed as 1x104.

In 2.3 line 91....500 uM?

Corrected to 500ul

- should specify the meaning of the abbreviation CLO;

Error, corrected to CBE

-should specify how normalization was performed for western blot analysis;

Corrected, information about normalization were added to Materials and methods section as follows: Western blot bands were densitometric analysed by Image Studio Lite software using β-actin as loading protein control. Untreated cell lysates was afterwards used to normalise all experimental groups.

-should insert in the legend of figure 2 ***;

Corrected; now Fig. 3

- should change the title of the results section 2.7 because they do not evaluate the activity. Superoxide anion is a ROS

corrected

Reviewer 2 Report

It is no presented clear information related to the composition and the stability of the tested extract.

In spite of the encouraging experimental data, the main limitation of the study is related to the testing on only one cell line.

A mechanistic chart, related to the biological active effect of CBE, from the ROS activity, to MAPK kinase, PI3K/Akt and DNA damage, emphasis the phosphorilathion of the proteins (if the case)

Author Response

Reviewer 2

It is no presented clear information related to the composition and the stability of the tested extract.

Composition of extract were detailed analysed (GC-MS and LC-MS-DAD analysis) in our previous paper (reference 12 as described in manuscript):

Kubatka, P., Uramova, S., Kello, M., Kajo, K., Kruzliak, P., Mojzis, J., ... & Zubor, P. (2017). Antineoplastic effects of clove buds (Syzygium aromaticum L.) in the model of breast carcinoma. Journal of cellular and molecular medicine, 21(11), 2837-2851.

In spite of the encouraging experimental data, the main limitation of the study is related to the testing on only one cell line.

It is true but we mentioned and planned these manuscript as upgrade of our above mentioned work. In Kubatka and col. paper we describe several preliminarily in vitro results and various in vivo results in breast carcinoma model (MCF-7 cell line and rats). In the present manuscript we described more detail of possible mechanisms mediated by CBE treatment. Therefore we chose same breast carcinoma cell line MCF-7. More cell lines will destruct consistency with our previous works. However, for comparison of cytotoxicity in cancer and non-cancer cell lines, we added also results of experiments on non-cancer breast cell MCF-10A (see Suppl 1)

A mechanistic chart, related to the biological active effect of CBE, from the ROS activity, to MAPK kinase, PI3K/Akt and DNA damage, emphasis the phosphorilathion of the proteins (if the case)

Correct, most of CBE-mediated effects are associated with phosphorylation of several proteins interconnected directly or indirectly in signalling cascade.

Reviewer 3 Report

After reviewing the manuscript entitled “Oxidative stress-induced DNA damage and apoptosis in clove buds-treated MCF-7 cells”, some modifications must be taken into account.

The manuscript the topic is well chosen, while several errors lead to consider the manuscript suitable for publication only after major revision.

I would recommend writing the article in third person. Therefore, review all the manuscript and make changes to eliminate “we”, “our” or “us” … Check the abstract and entire manuscript and keep consistency.

Use the International Code of Nomenclature for plants, adding the authority after the binomial name and the family.

Material and methods:

As the authors guide says “They should be described with sufficient detail to allow others to replicate and build on published results. --- well-established methods can be briefly described and appropriately cited.” This part should be described a little bit more, and REFERENCED. There is no reference in the methods.

Also, “Give the name and version of any software used”, especially for statistical analysis.

Separate the reagents (seller and origin) from the methods and include them in the reagents/material section.

Moreover, in this section must be completed the information about the plant and the extract. Where was it collected / purchased? In addition, whenever working with products of natural origin, it is advisable to save a specimen (name the identification code if applicable), or if it has been purchased, specify the seller and origin. You must also define which parts of the plants were used to make the extract, the amount of plant and solvent used, as well as the yield of the preparation.

Results and discussion:

The figures should not be framed... They should be improved by increasing the font size, especially fig 2 and 4c. I would attach them one by one in order to later be able to make up a better manuscript.

Does the control have no deviations? if different controls are not carried out, it is easy to obtain significant differences later ...

I hope that the protein expression analysis is normalized with actin (Densitometry analyses of western blot results), but it is not mentioned anywhere in the manuscript.

The extract should be phytochemically characterized, at least in total polyphenols, flavonoids, essential oil... or reference correctly previous works with the SAME extract plant.

To improve it, a deeper discussion must take place; I would also include in discussion section the activities already seen in others papers by the isolated phytochemicals present in the extract.

Author Response

Reviewer 3

I would recommend writing the article in third person. Therefore, review all the manuscript and make changes to eliminate “we”, “our” or “us” … Check the abstract and entire manuscript and keep consistency.

Partial accepted. It is not possible to eliminate “we”, “our” or “us” when we want discuss our data published in our previous papers. And we want tell readers which data was made by our research group. This article is research article and we want take all our results in broader context as possible (including results from our previous papers if relative).

Use the International Code of Nomenclature for plants, adding the authority after the binomial name and the family.

Corrected

Material and methods:

As the authors guide says “They should be described with sufficient detail to allow others to replicate and build on published results. --- well-established methods can be briefly described and appropriately cited.” This part should be described a little bit more, and REFERENCED. There is no reference in the methods.

Based on our assessment, we described methods used in our experiment detailed with all necessarily data needed to repeat experiments (we corrected now some parts). References was added to method section.

Also, “Give the name and version of any software used”, especially for statistical analysis.

Corrected: Data analyses were conducted using GRAPHPAD PRISM, version 5.01 (GraphPad Software, La Jolla, CA, USA).

Separate the reagents (seller and origin) from the methods and include them in the reagents/material section.

Accepted, Reagents section (2.2.) was added and seller origin was removed from text.

Moreover, in this section must be completed the information about the plant and the extract. Where was it collected / purchased? In addition, whenever working with products of natural origin, it is advisable to save a specimen (name the identification code if applicable), or if it has been purchased, specify the seller and origin. You must also define which parts of the plants were used to make the extract, the amount of plant and solvent used, as well as the yield of the preparation.

These data are already presented in manuscript section 2.1. We renamed section to more precise description as: Cell cultures and CBE treatment

Results and discussion:

The figures should not be framed... They should be improved by increasing the font size, especially fig 2 and 4c. I would attach them one by one in order to later be able to make up a better manuscript.

All figures were loaded separately in manuscript uploading process as png/jpg files. Based on the instruction for authors, in process of making manuscript, the figures should include also in text in place we want and for review process. After acceptance of manuscript, we are prepared to assist to production staff if some changes in figures are needed to improve quality or if separation of graphs will be necessary.

Does the control have no deviations? if different controls are not carried out, it is easy to obtain significant differences later ...

Correct, data presented are normalised to untreated control group which have set value to 1 (or 100%). We compare raw data with control group data and made normalisation. We take data from 3 independent experiments and made averages. Therefore average from 3x value 1 is zero and control have DV=0. Flow cytometry data cannot be simple averaged as raw data because living system can show bigger variation between weeks of experiments but normalised data showed clearly consistency of repeats.

I hope that the protein expression analysis is normalized with actin (Densitometry analyses of western blot results), but it is not mentioned anywhere in the manuscript.

Accepted. All protein expression analysis is normalized with actin and information about them was added to method section. Western blot bands were densitometricaly analysed by Image Studio Lite software.

The extract should be phytochemically characterized, at least in total polyphenols, flavonoids, essential oil... or reference correctly previous works with the SAME extract plant.

These data are already presented in manuscript- reference 12 (end of the section 2.1)

To improve it, a deeper discussion must take place; I would also include in discussion section the activities already seen in others papers by the isolated phytochemicals present in the extract.

To the best of our knowledge, this is the first study reporting association of Erk, JNK and p38 MAPK in ROS-mediated apoptosis after clove buds extract treatment in carcinoma model. But other presented data we discussed mostly with eugenol or flavonoids/polyphenols compounds, if exist.

Round 2

Reviewer 1 Report

after the revision the manuscript has improved

Author Response

We would like to thank the reviewer for his/her comments, which have given us the opportunity to improve the manuscript.

Reviewer 2 Report

The paper can be accepted after adding a mechanistic chart related to the biological effect of clove buds extract.

Author Response

We would like to thank the reviewer for his/her comments, which have given us the opportunity to improve the manuscript.

Mechanistic chart related to the biological effect of clove buds extract (Fig. 6) has been added to the end of Discussion.

Reviewer 3 Report

The changes made have improved the manuscript considerably, however some changes should be made.

It's completely possible eliminate we, us etc. our previous papers → Previous articles of the authors ...

The preparation of the extract is not clear. Do you buy the extract? the plant? in the methodology section include the part of the plant (clove buds ??), because only powder is said ...   If it's a maceration ... for how long? How was the solvent removed? This section has not been changed and should be improved.

To avoid having zero deviations, several controls must always be done in the same experiment ... because otherwise, the significant differences with respect to three values without deviation are not true… it's a serious mistake.

Author Response

We would like to thank the reviewer for his/her comments, which have given us the opportunity to improve the manuscript.

1. It's completely possible eliminate we, us etc. our previous papers → Previous articles of the authors ...

accepted; corrected in the text

2. The preparation of the extract is not clear. Do you buy the extract? the plant? in the methodology section include the part of the plant (clove buds ??), because only powder is said ... If it's a maceration ... for how long? How was the solvent removed? This section has not been changed and should be improved.

accepted; described in the chapter 2.1.

3. To avoid having zero deviations, several controls must always be done in the same experiment ... because otherwise, the significant differences with respect to three values without deviation are not true… it's a serious mistake.

see PDF file
